# Economic evaluation of antimicrobial resistance in curable sexually transmitted infections; a systematic review and a case study

Oluseyi Ayinde[1], Jonathan D. C. Ross[1], Louise Jackson[2]*

1 Sexual Health and HIV, University Hospitals Birmingham NHS Trust, Birmingham, United Kingdom,
2 Health Economics Unit, Institute of Applied Health Research, College of Medical and Dental Sciences, University of Birmingham, Birmingham, United Kingdom

* l.jackson.1@bham.ac.uk

## Abstract

### Objective

To provide a summary of the economic and methodological evidence on capturing antimicrobial resistance (AMR) associated costs for curable sexually transmitted infections (STIs). To explore approaches for incorporating the cost of AMR within an economic model evaluating different treatment strategies for gonorrhoea, as a case study.

### Methods

A systematic review protocol was registered on PROSPERO (CRD42022298232). MEDLINE, EMBASE, CINAHL, Cochrane Library, International Health Technology Assessment Database, National Health Service Economic Evaluation Database, and EconLit databases were searched up to August 2022. Included studies were analysed, quality assessed and findings synthesised narratively. Additionally, an economic evaluation which incorporated AMR was undertaken using a decision tree model and primary data from a randomised clinical trial comparing gentamicin therapy with standard treatment (ceftriaxone). AMR was incorporated into the evaluation using three approaches—integrating the additional costs of treating resistant infections, conducting a threshold analysis, and accounting for the societal cost of resistance for the antibiotic consumed.

### Results

Twelve studies were included in the systematic review with the majority focussed on AMR in gonorrhoea. The cost of ceftriaxone resistant gonorrhoea and the cost of ceftriaxone sparing strategies were significant and related to the direct medical costs from persistent gonorrhoea infections, sequelae of untreated infections, gonorrhoea attributable-HIV transmission and AMR testing. However, AMR definition, the collection and incorporation of AMR associated costs, and the perspectives adopted were inconsistent or limited. Using the review findings, different approaches were explored for incorporating AMR into an economic

**Funding:** The author(s) received no specific funding for this work.

**Competing interests:** JDCR reports personal fees from GSK Pharma and Hologic Diagnostics, as well as ownership of shares in GSK Pharma and Astrazeneca Pharma; and is author of the UK and European Guidelines on Pelvic Inflammatory Disease; is a Member of the European Sexually Transmitted Infections Guidelines Editorial Board. He is an NIHR Journals Editor and associate editor of Sexually Transmitted Infections journal. He is an officer of the International Union against Sexually Transmitted Infections (treasurer). LJ and OA report no conflicts of interest. This does not alter our adherence to PLOS ONE policies on sharing data and materials.

evaluation comparing gentamicin to ceftriaxone for gonorrhoea treatment. Although the initial analysis showed that ceftriaxone was the cheaper treatment, gentamicin became cost-neutral if the clinical efficacy of ceftriaxone reduced from 98% to 92%. By incorporating societal costs of antibiotic use, gentamicin became cost-neutral if the cost of ceftriaxone treatment increased from £4.60 to £8.44 per patient.

## Conclusions

Inclusion of AMR into economic evaluations may substantially influence estimates of cost-effectiveness and affect subsequent treatment recommendations for gonorrhoea and other STIs. However, robust data on the cost of AMR and a standardised approach for conducting economic evaluations for STI treatment which incorporate AMR are lacking, and requires further developmental research.

## Introduction

Chlamydia, gonorrhoea, syphilis, and trichomoniasis are the most prevalent sexually transmitted infections (STIs) globally, with a reported incidence of 374 million infections in 2020 [1]. Though mostly curable, many of these infections are becoming increasingly resistant to first-line treatments [2, 3]. For instance, gonorrhoea, the world's second most prevalent bacterial STI has progressively developed resistance over the last 80 years to a wide variety of antibiotic regimens including sulfonamides, penicillins, tetracyclines, macrolides, and fluoroquinolones [4]. In many countries, first line therapy for gonorrhoea is limited to extended-spectrum cephalosporins, such as cefixime and ceftriaxone [5]. Consequently, the World Health Organization (WHO) has identified gonorrhoea as one of the top 12 priority pathogens for new antibiotic research and development [6].

The pipeline for new drugs to treat emerging resistance is limited due to high development costs and challenges around formulation, regulation, and profitability [7]. Therefore, there is an urgent need for strategies to limit the development of antimicrobial resistance (AMR). These include the use of existing 'older' antibiotic treatments (where possible) or changes in care pathways to slow the spread of AMR. A number of studies have assessed the efficacy of alternative antibiotics [8, 9], and explored the use of AMR guided therapy and the role of antibiotic stewardship [10–12]. However, there is little evidence regarding the costs associated with AMR for curable STIs and how this affects the cost-effectiveness of interventions designed to reduce AMR [13–16]. This limits comprehensive evaluation of AMR control strategies and the cost implications for patient management, which are important to clinicians, commissioners and policy makers when developing new management guidance [17, 18]. In addition, data on the economic impact of AMR is needed to direct targeted investment into future drug development.

Therefore, we (i) performed a systematic review to appraise the economic evidence relating to AMR for curable STIs, and provide a comprehensive overview of the methods currently used to incorporate AMR into economic evaluations of treatments in patients with curable sexually transmitted infections, and (ii) used a case study to explore how AMR could be incorporated into an economic evaluation using a decision-analytic model incorporating data from a large pragmatic multicentre randomised clinical trial (RCT) which recruited patients with gonorrhoea.

## Methods

### Systematic review

A systematic review was conducted following the guidelines of the Centre for Review and Dissemination [19] and the Preferred Reporting Items for Systematic Reviews and Meta-Analyses (PRISMA) [20] to answer the research question—What evidence is available on the costs associated with AMR in curable STIs and what methods have been adopted to include such costs in economic evaluations? A systematic review protocol was developed and registered with PROSPERO at the CRD, University of York (Registration No CRD42022298232)— https://www.crd.york.ac.uk/prospero/display_record.php?ID=CRD42022298232.

**Eligibility criteria.** Studies were considered eligible for review if they met the following criteria:

i. P–the population consisted of men or women of any age with a curable STI (gonorrhoea, chlamydia, syphilis, trichomoniasis, *Mycoplasma genitalium*)

ii. I–the intervention was treatment of the STI

iii. C–the comparator was licensed or unlicensed pharmacological or non-pharmacological treatments of the STI

iv. O–outcomes were reported in terms of costs, economic evaluations and health outcomes. We also evaluated approaches to modelling, modelling assumptions and proxy outcomes associated with AMR.

There was no restriction to the study setting or the publication date. However, publications were restricted to those in the English language.

**Search strategy.** Scoping searches were initially carried out to refine the search strategy. Thereafter, the following databases were searched from inception to August 2022. MEDLINE, EMBASE, British Nursing Index and Cumulative Index Nursing and Allied Health Literature, Cochrane Library, International Health Technology Assessment Database, National Health Service Economic Evaluation Database, and EconLit. In addition, relevant websites related to STIs and economic evaluations were searched; the National Institute for Health and Care Excellence, Cost-Effectiveness Analysis (CEA) Registry, Research Papers in Economics, WHO, Public Health England, Centers for Disease Control and Prevention (CDC), the European Centre for Disease Prevention and Control, and the British Association for Sexual Health and HIV. Reference lists of included selected studies were hand searched. The database search strategy is reported in S1 File.

**Study selection.** All identified records were transferred to EndNote referencing manager (V.X9) for management and categorisation. A two-stage process as outlined by Roberts [21] was used to select studies. In stage I, titles and abstract were screened and assigned into categories A to G. In stage II, full text articles of studies categorised A to C, were further categorised into 1–8 (S2 File). The identification and initial categorisation were performed by two reviewers. A third reviewer checked subsets of the selection process (screening, eligibility and inclusion) to confirm the categorisation of studies. Studies categorised A to C and grouped into 1 to 3 were included in the review (Fig 1).

**Data extraction and synthesis.** Data extraction was performed independently by two reviewers on all included studies. Disagreements were resolved by discussion and by a third independent reviewer. The extracted data were tabulated (data extraction form, S1 Table) and synthesised narratively. A narrative synthesis was adopted this being the most appropriate approach for bringing together studies with heterogeneous methodologies [19].

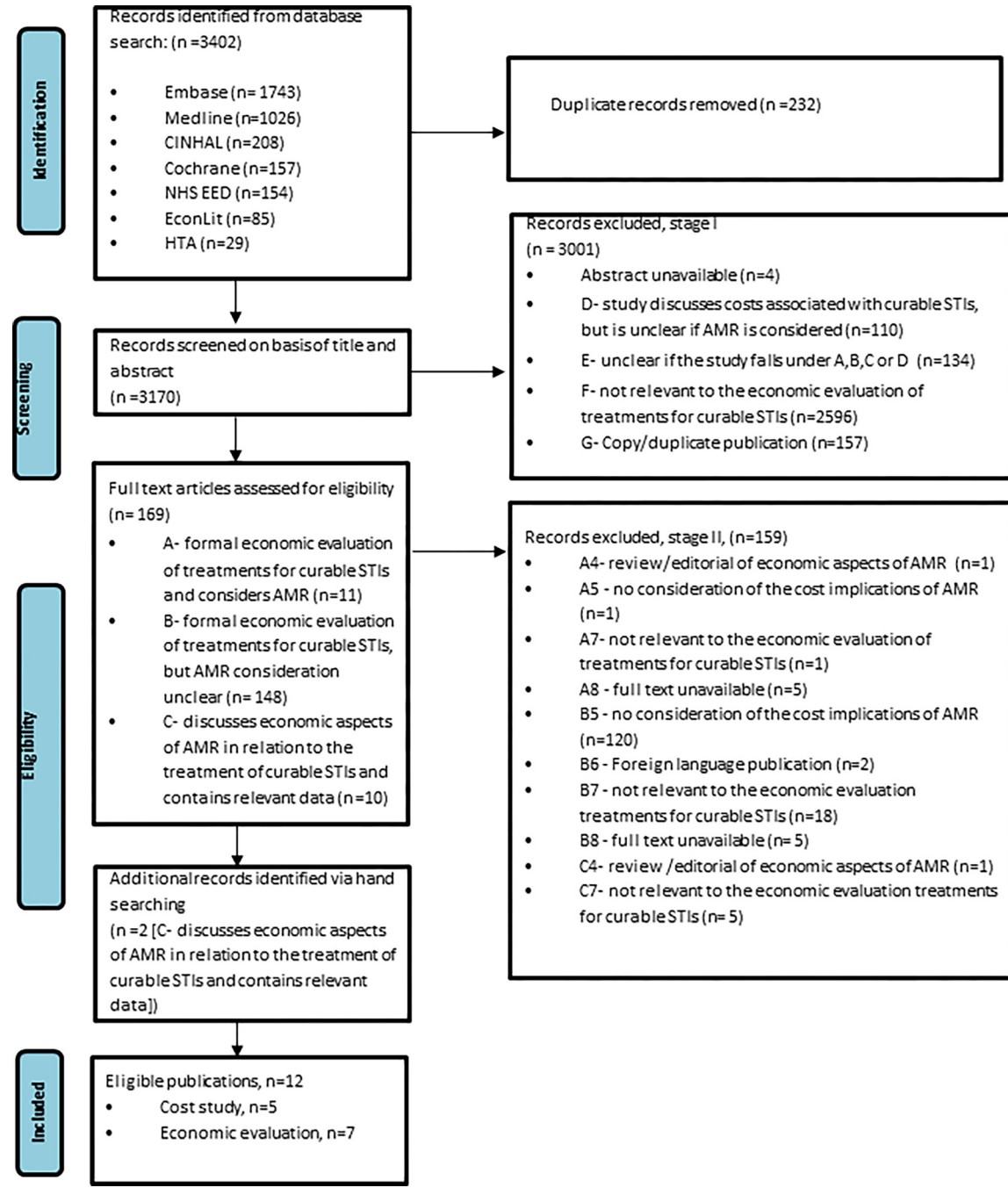

**Fig 1. PRISMA flow diagram.**

**Quality assessment.** The quality of included studies was assessed using an adapted version of the Consolidated Health Economic Evaluation Reporting Standards (CHEERS) checklist [22] and the checklist of cost of illness [23] for economic evaluations and cost studies respectively (S3 File). Quality assessment was used to inform the synthesis but no studies were excluded on the basis of quality.

## Economic evaluation

For the second element of this study, we used the findings of the systematic review to explore different methods to incorporate the costs associated with AMR within an economic evaluation. For this component we adapted an existing model developed for a RCT concerned with different treatment strategies for gonorrhoea. The methods and results of the 'Gentamicin compared with ceftriaxone for the treatment of gonorrhoea RCT (GToG)' RCT are reported elsewhere [24]. In brief, a blinded, non-inferiority RCT was conducted in 14 sexual health clinics in England. 720 adult sexual health clinic attendees with uncomplicated gonorrhoea were randomised 1:1 to receive either gentamicin 240 mg or ceftriaxone 500 mg, both administered as a single intramuscular injection. All participants also received 1 g oral azithromycin. The primary outcome was clearance of *Neisseria gonorrhoeae* at all initially test positive sites two weeks after treatment. The GToG RCT was approved by Health Research Authority South Central–Oxford C Research Ethics Committee (14/SC/1030). The trial was registered prior to start of recruitment (ISRCTN51783227).

**Model structure.**   A simple decision tree model was developed using TreeAge Pro 2016 (TreeAge Software Inc., Williamstown, MA, USA). The structure was informed by the trial objectives and patient pathways indicated by the clinical data. Patients entered the model at the point of randomisation when they were assigned to receive the alternative treatment (gentamicin) or the standard treatment (ceftriaxone). Following the initial course of antibiotic treatment, patients either received additional NHS care (e.g. General Practitioner—GP visit) or did not access care. At two weeks post-treatment, patients were either cleared of *Neisseria gonorrhoeae* (confirmed by a negative nucleic acid amplification test [NAAT]) or they were not cleared and required further treatment (Fig 2). The economic analysis focused on establishing if gentamicin compared to ceftriaxone was cost neutral in the treatment of gonorrhoea, which was deemed to be most relevant for a non-inferiority trial. The analysis was undertaken from the perspective of the health service (NHS). For the initial economic evaluation, the time horizon was two weeks and restricted to the follow-up time within the RCT.

**Data sources.**   Clinical data on the primary outcome (microbial resolution), resource use and costs were collected prospectively via trial reporting mechanisms (Tables 1 and 2). Data on additional resource use after initial treatment and prior to the 2-week test of cure was also collected prospectively (Table 3). Additional data relating to unit costs and scenarios on further treatment due to non-clearance of infection at the 2-week test of cure were sourced from the literature (Table 4).

**Initial analysis.**   The cost per patient successfully treated (measured in terms of microbial clearance of *Neisseria gonorrhoeae* at all infected sites) was estimated. As the trial was concerned with the immediate post-treatment period (two weeks), discounting was not undertaken. All costs are given in £UK for 2020/2021. A probabilistic sensitivity analysis was undertaken to explore uncertainties. All parameters were varied simultaneously sampling multiple sets of parameter values from defined probability distributions. A Monte Carlo simulation was used to sample from the distributions; this involved 1000 repeated random draws to analyse how variation in the parameters used in the model would affect the results. For binomial data, beta distributions were used, and gamma distributions were used for costs, in line with recommendations for specifying distributions for parameters [27].

**Secondary analysis—accounting for antimicrobial resistance.**   Building on the results of the systematic review of the literature, we employed three approaches to account for AMR— (i) including the additional costs associated with treating resistant infections [13], (ii) using a threshold analysis to assess the level of resistance required to impact on cost to potentially change the decision [15], and (iii) estimating and including the societal cost of resistance for

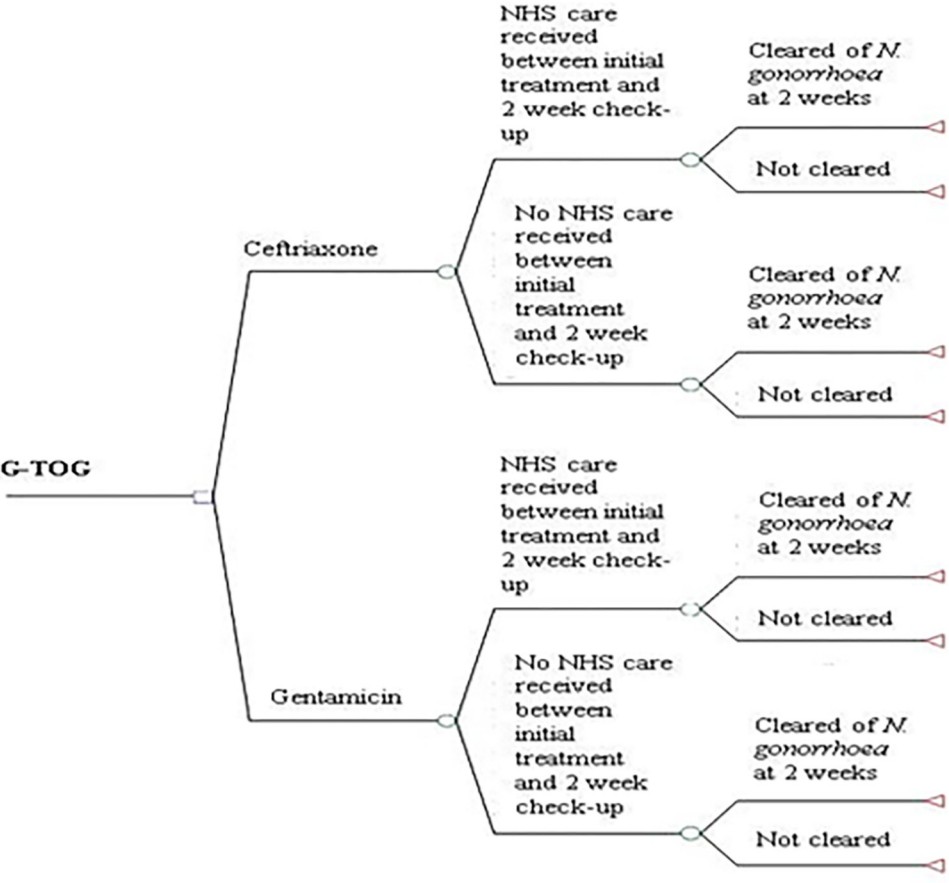

**Fig 2. Initial model structure.**

each antibiotic consumed [15, 18]. These approaches were analysed in the context of alternative antibiotic treatments for gonorrhoea.

i. Additional costs associated with treating resistance to ceftriaxone: We identified an appropriate treatment pathway which would be followed if an infection was resistant to ceftriaxone (Fig 3). This was informed by current and proposed guidelines for the treatment of gonorrhea at the time [28, 29]. The additional cost included cost of additional clinic visits,

**Table 1. Probabilities used in decision tree model.**

| Description | Trial data | Probability | Distribution |
|---|---|---|---|
| **Ceftriaxone arm** | | | |
| Requiring NHS treatment after the initial visit | 10/322 | 0.03 | Beta |
| Not requiring NHS treatment after the initial visit | 312/322 | 0.97 | Beta |
| Clearance of *N. gonorrhoeae* | 299/306 | 0.98 | Beta |
| Not cleared *N. gonorrhoeae* | 7/306 | 0.02 | Beta |
| **Gentamicin arm** | | | |
| Requiring NHS treatment after the initial visit | 8/302 | 0.03 | Beta |
| Not requiring NHS treatment after the initial visit | 294/302 | 0.96 | Beta |
| Clearance of *N. gonorrhoeae* | 267/292 | 0.91 | Beta |
| Not cleared *N. gonorrhoeae* | 25/292 | 0.09 | Beta |

**Table 2. Trial treatments.**

| Resource use | Cost item | Base case value (£) | Distribution | Source |
|---|---|---|---|---|
| Gentamicin treatment | Per patient | 3.60[1] | Gamma | BNF |
| Ceftriaxone treatment | Per patient | 4.60[2] | Gamma | BNF |

1 This was from three 80 mg ampoules, with costs estimated at £1.20 per ampule (£3.60 overall, British National Formulary [BNF]) [25]

2 Patients allocated to the ceftriaxone arm received a 500 mg dose. Ceftriaxone was purchased in units of 1 g and mixed with 4 ml (1%) lidocaine solution. Only half of the preparation was administered to the patient (half was discarded), we therefore included costs for one vial of 1 g ceftriaxone powder (£3.60) per patient and 2 × 2 ml lidocaine ampule (50p per ampule [source BNF]) [25]

and additional courses of antibiotics (following guidance, we assumed that gentamicin would be used as treatment if the infection was resistant to ceftriaxone) (Table 5).

ii. Threshold analysis on the level of ceftriaxone resistance necessary for gentamicin to be cost-neutral: this involved varying the value of a parameter that was important to the analysis and assessing what value estimate would cause a programme or intervention to be cost-effective or not cost-effective (or cost-neutral in this case) [30]. The level of gonorrhoea ceftriaxone resistance was varied as this was judged to be critical and would affect the context in which decision-making around treatment takes place.

iii. Including the societal cost of resistance: We applied the societal costs associated with AMR to the ceftriaxone treatment arm only. This is because the extended-spectrum cephalosporins, cefixime and ceftriaxone are the only remaining monotherapy that can be used effectively to treat gonorrhoea. Therefore, in reality the costs associated with resistance in ceftriaxone would be higher for society. The costs associated with ceftriaxone resistance for each course of antibiotic consumed as reported by Shrestha et al. [18] ranged between $1 and $25.6. These estimates were converted to UK pounds and inflated.

For this secondary analysis, gonorrhea AMR was defined as persistent gonorrhea infection determined by a positive gonorrhea NAAT test at the 2-week test of cure at anatomical sites

**Table 3. NHS Resource use–after initial treatment and before the two week check-up.**

| Resource use | Cost item | Unit cost (UK £) | N Ceftriaxone Group | N Gentamicin Group | Total cost- Ceftriaxone (UK £) | Total cost–Gentamicin (UK £) |
|---|---|---|---|---|---|---|
| GP consultation | Per visit | 39.23 | 6 | 3 | 235.38 | 117.69 |
| Sexual Health—health advisor consultation | Per visit | 78* | 1 | 2 | 78 | 156 |
| Sexual health clinic–doctor consultation | Per visit | 124** | 5 | 4 | 620 | 496 |
| NHS 111 calls | Per call | 7.62 *** | 1 | 1 | 7.62 | 7.62 |
| Total costs | | | | | 941 | 777.31 |
| Total number of patients accessing additional treatment | - | - | 10 | 8 | - | - |
| Total cost per patient accessing additional treatment | - | - | - | - | 94.10 | 97.16 |

* Assumes equivalent to non-consultant led family planning clinic–outpatient attendance, PSSRU 2020/21

** Assumes equivalent to consultant-led family planning clinic consultation–outpatient attendance, PSSRU 2020/21

*** Assumes equivalent to nurse-led telephone consultation, PSSRU 2020/21 [26].

GP- General Practitioner

**Table 4. Costs of further treatment for patients where infection was not cleared.**

| Resource use | Cost item | Unit cost (£) | Source |
|---|---|---|---|
| Sexual Health Centre–nurse/health advisor consultation | Per visit | 78* | PSSRU 2020/21 |
| Second course of antibiotic treatment | Per patient | 4.60** | BNF |
| Total costs | | 82.60 | |

*Assumes equivalent to non-consultant led family planning clinic–outpatient attendance

** Assumes 1 injection of ceftriaxone.

that were previously tested. Further assumptions made for the initial and secondary analyses are reported in S4 File.

## Results

### Systematic review findings

**Study selection.** 3,402 studies were identified from the database search and 232 duplicate records were removed. Using the 2-stage process to screen and identify eligible studies, in Stage I titles and abstracts of 3170 records were screened for relevance, of these, 3001 articles were excluded and 169 relevant full-text articles were further assessed for eligibility in Stage II. Following full text analysis, 159 were excluded and 12 full text articles—10 identified from the database search and an additional 2 full text articles identified by hand searching were included in the review (Fig 1).

**Study characteristics.** The characteristics of the included studies are summarised in Table 6. Overall, 7/12 studies were economic evaluations of which most employed a cost-effectiveness design (6/7). 5/12 studies were cost studies including a national action plan and cost-analysis, cost minimisation and cost modelling studies. The study population was heterogeneous, including adolescents and adults, men and women, national populations and specific groups such as sex workers, pregnant women or men who have sex with men (MSM). 7/12 included studies which originated from the United States and 11/12 were specific to gonorrhoea infection.

The aims of the included studies varied, including estimating the costs or cost-effectiveness of AMR testing strategies (6/12), cost projections for emerging AMR (2/12), and evaluating interventions for STI management with AMR as an adjunct to the analysis (4/12). Almost all of the included studies (11/12) reported only direct medical costs (Table 6).

**Review of the cost of AMR.** The economic consequences of AMR identified from the review can be presented in four main themes—(i) the projected cost of emerging gonorrhoea

- Initial clinic visit & treatment with ceftriaxone
- Follow up clinic visit - positive NAAT (test of cure)
- Additional antibiotic (ceftriaxone) & sensitivity test
- Additional follow up clinic visit – further sensitivity test
- Additional antibiotic (gentamicin + azithromycin)

**Fig 3. Scenario for treatment of gonorrhoea resistant to ceftriaxone.**

**Table 5. Additional cost for treatment of strain resistant to ceftriaxone (excluding initial clinic visit).**

| Resource use | Cost item | Unit cost (£) | Source |
|---|---|---|---|
| 2x Sexual Health Centre–nurse/health advisor consultation | Per visit | 78* | PSSRU 2020/21 |
| Second course of antibiotic treatment (ceftriaxone) | Per patient | 4.60** | BNF |
| Third course of antibiotic treatment (gentamicin) | Per patient | 3.60*** | BNF |
| Total costs | Per patient | 164.20 | - |

*Assumes equivalent to non-consultant led family planning clinic–outpatient attendance

** Assumes 1 injection of ceftriaxone

*** Assumes 1 injection of gentamicin

cephalosporin resistance, (ii) cost of dual therapy in managing gonorrhoea AMR, (iii) cost associated with AMR testing and targeted antibiotic treatment and (iv) threshold parameters associated with AMR testing.

i.  *Costs of emerging gonorrhoea AMR to cephalosporin*: Two cost studies from the United States found the cost of emerging gonorrhoea AMR to first-line cephalosporin therapy over a 10-year period could be substantial (Table 7). Cost projections from the CDC suggests that widespread gonorrhoea cephalosporin resistance resulted in additional cases of pelvic inflammatory disease, epididymitis and HIV infections with a cumulative direct medical cost of at least $235 million [36]. Similarly, Chesson et al. [38] estimated that emerging gonorrhoea ceftriaxone resistance (rising from 2% to 15%) could cost $378.2 million, resulting from 1,157,100 additional gonorrhoea infections and 579 gonorrhoea-attributable HIV infections (Table 7).

ii.  *Costs of dual therapy in managing gonorrhoea AMR*: In response to the evolving gonorrhoea ceftriaxone resistance, some guidelines recommend a dual antibiotic treatment regimen [41]. Xiridou et al. [37] investigated the cost-effectiveness of dual therapy (ceftriaxone plus azithromycin) compared to ceftriaxone monotherapy using a transmission model, and found that dual therapy slowed down the spread of resistance (5% resistance threshold) by at least 15 years in MSM, but resulted in additional treatment cost over 60 years (cumulative ICER- cost/QALY of €9.74 x $10^8$ and €14866 for 10 and 60 years respectively) (Table 8). The analysis was based on the cost of medical consultation, testing, treatment, quality of life loss for symptomatic gonorrhoea, duration of infection and 10% prevalence of gonorrhoea complications in symptomatic patients. The cost-effectiveness estimate was sensitive to the costs of consultations, tests and the weighting for quality of life loss. When there was initial azithromycin resistance (5%), dual therapy was not cost-effective and unlikely to preserve ceftriaxone use.

iii.  *Cost of gonorrhoea AMR testing and targeted antibiotic treatment*: The costs of AMR testing and antibiotic stewardship was investigated in five studies. Three studies [11, 12, 39] compared AMR testing (point-of-care-test POCT and AMR screening) to improve antibiotic stewardship with dual therapy (ceftriaxone plus azithromycin) for gonorrhoea. These studies consistently found that AMR testing and treatment cost more compared to dual therapy (Table 8). In one study, the additional cost for optimal treatment (using a ceftriaxone sparing regimen) was at least £414.7, and all strategies which involved dual resistance testing (ciprofloxacin and azithromycin) were dominated by standard of care. The results were associated with a base case cost of £29.00 and £31.90 for single and dual resistance testing respectively [39]. Similarly, Turner et al. [11] found gonorrhoea AMR POCT (gonorrhoea ciprofloxacin and/or penicillin resistance) led to an additional cost of £34 million per

**Table 6. Characteristics of included studies (by year).**

| Author | Year | Country | Study type | Design | Study aim/objective | Study population | STI | Type of Cost (Currency) |
|---|---|---|---|---|---|---|---|---|
| Phillips et al. [31] | 1989 | United States | Economic evaluation | CEA model | To calculate the economic implications of routine testing for NG infection of the cervix. | Women aged (18–40 years) attending for routine gynaecologic care. | NG | Direct medical cost and indirect costs (USD)[1] |
| Nettleman et al. [32] | 1990 | United States | Economic evaluation | CEA model | To investigate the cost effectiveness of beta-lactamase screening and alternative therapies for patients attending STD clinics in low prevalence areas. | Patients attending sexual transmitted diseases clinics. | NG | Direct medical costs (USD) |
| Crabbe et al. [33] | 2000 | Multi-national (developing world) [2] | Economic evaluation | CEA model | To recommend a cost-effective approach for the management of acute male urethritis in the developing world based on the findings of a theoretical study. | Male patients with signs/symptoms of urethritis. | NG/ NGU | Direct medical costs (USD) |
| Roy et al. [34] | 2005 | United States | Economic evaluation | CEA model | To identify the most cost-effective combination of diagnostic test (culture with antimicrobial susceptibility versus nonculture) and treatment (ciprofloxacin versus ceftriaxone) for gonorrhoea when the incidence of ciprofloxacin-resistant NG infections is increasing. | Women | NG | Direct medical costs (USD) |
| Price et al. [35] | 2006 | Malawi | Economic evaluation | CEA model | To examine the cost-effectiveness of providing first-line treatment for male trichomoniasis in Malawi. | Semi-urban men seeking STD treatment. | TV | Direct medical costs (USD) |
| CDC [36] | 2015 | United States | Cost study | National action Plan | To outline steps for implementing the National strategy for combating antibiotic-resistant bacteria and addressing the policy recommendations of the President's council of Advisors on Science and Technology. | United States population. | NG | Direct medical cost (USD) |
| Xiridou et al. [37] | 2016 | Netherlands | Economic evaluation | CUA model | To investigated the cost-effectiveness of dual therapy with ceftriaxone and azithromycin compared with monotherapy with ceftriaxone, for control of NG among men who have sex with men in the Netherlands. | MSM receiving treatment for NG. | NG | Direct medical cost (Euros) |
| Turner et al. [11] | 2017 | United Kingdom | Cost study | Cost Modelling | To create a mathematical model to investigate the treatment impact and economic implications of introducing an AMR POCT for gonorrhoea as a way of extending the life of current last-line treatments. | Patients accessing sexual health services. | NG | Direct medical costs (GBP) |
| Allan-Blitz et al. [12] | 2018 | United States | Cost study | Cost analysis | To compare the actual costs of an ongoing program for *gyrA* genotyping and targeted ciprofloxacin therapy at the University of California, Los Angeles over a thirteen-month period with the costs of recommended two drug ceftriaxone and azithromycin therapy. | Patients diagnosed of gonorrhoea infection. | NG | Direct medical cost (USD) |
| Chesson et al. [38] | 2018 | United States | Cost study | Cost Modelling | To perform a modelling exercise of an illustrative scenario of increased gonorrhoea incidence in the United States caused by emerging cephalosporin resistance. | Annual NG infections in United States. | NG | Direct medical cost (USD) |
| Harding-Esch et al. [39] | 2020 | United Kingdom | Economic evaluation | CEA model | To assess the costs and effectiveness of AMR POCT strategies that optimises NG treatment and reduces ceftriaxone use. | Sexual health clinic attendees diagnosed of NG. | NG | Direct medical cost (GBP/ Euros) |

(*Continued*)

**Table 6.** (Continued)

| Author | Year | Country | Study type | Design | Study aim/objective | Study population | STI | Type of Cost (Currency) |
|---|---|---|---|---|---|---|---|---|
| Wynn and Klausner [40] | 2020 | United States | Cost study | CMA | To identify the price point at which the additional cost of a *gyrA* assay to NG management would either break even or generate cost savings compared with the current standard of care in the US. | Asymptomatic individuals seeking STI screening. | NG | Direct medical cost (USD) |

1. Indirect cost represents loss of wages, loss of household management due to disability, or loss of lifetime earnings due to death.

2. Burkina Faso, Congo, Ghana, Mali, Chad, Ecuador, Haiti, Nicaragua, Bangladesh, Cambodia, China, Laos, and Vietnam.

AMR, antimicrobial resistance; CDC, Centers for Disease Control and Prevention; CEA, cost effectiveness analysis; CMA, cost minimization analysis; CT, chlamydia; CUA, cost utility analysis; GBP, Great British Pound; NG, gonorrhoea; NGU, non-gonococcal urethritis; POCT, point-of-care test; STI-sexually transmitted infection; STD- sexually transmitted disease; Syp, syphilis; TV- trichomoniasis; US- United States; USD-US dollars.

annum compared to the standard of care, assuming AMR POCT added £25 to the first-line testing cost and that 66% of ceftriaxone treatments could be replaced by ciprofloxacin annually. A screening test for gonorrhoea ciprofloxacin resistance (assessing DNA gyrase —*gyrA*) followed by targeted therapy with ciprofloxacin or ceftriaxone plus azithromycin was estimated to cost an additional $54.40 (minimum $12.40) per patient compared to the standard of care (2-drug ceftriaxone and azithromycin therapy) in a retrospective study [12]. These costs were based on a test cost of $100.50 (test and labour), 35.0% prevalence of ciprofloxacin resistance, and 30.3% incidence of indeterminate genotype test results. In contrast, prior to the introduction of ceftriaxone as first line treatment for gonorrhoea treatment, gonorrhoea AMR testing and targeted therapy was found to be cost-effective [32, 34]. Beta-lactamase (an enzyme associated with resistance to ß-lactam antibiotics such as penicillin) screening and antibiotic stewardship was cost-effective compared to empirical treatment with ceftriaxone, assuming penicillin resistance at 5% and a $0.50 per ß-lactamase screening test [32]. Roy et al. [34] also found that culture-based testing strategies (strategies 1 and 3) were optimal (lowest cost per patient successfully treated) at lower levels of gonorrhoea prevalence ($\leq$ 5%)—Table 8.

iv. *Threshold parameters associated with gonorrhoea AMR testing*: Two studies reported on threshold analyses in relation to AMR. Wynn and Klausner [40] found the breakeven price for using a *gyrA* assay to detect ciprofloxacin resistance followed by targeted antibiotic treatment compared to standard dual therapy was $50 per *gyrA* test for the treatment of asymptomatic patients being screened for STIs, assuming a gonorrhoea prevalence of 2%, ciprofloxacin susceptibility of 70%, and *gyrA* assay sensitivity and specificity of 98 and 99%. The breakeven cost was sensitive to the prevalence of ciprofloxacin susceptibility, the cost of standard of care, and the frequency of indeterminant *gryA* test findings, but insensitive to gonorrhoea prevalence (2% or 8%). A threshold gonorrhoea prevalence of 2.5% was reported by Phillips et al. [31] for reducing the direct medical costs of gonorrhoea management in women when comparing routine gonorrhoea culture and susceptibility testing ($9 per culture test) with 'no' testing. However, the threshold prevalence was sensitive to the cost of culture tests and risk of adverse sequelae after treatment.

**Review of the methodological approaches for inclusion of AMR in economic evaluations.** The following key methodological elements and strategies were associated with incorporating AMR into cost/economic evaluations.

**Table 7. Methodological specifications of cost studies of AMR in the treatment of curable STIs (by year).**

| Author | Strategies/interventions/comparators | Source of Cost | Horizon (Discount rate) | Perspective | Definition of AMR | Method of AMR incorporation | AMR Assumption | AMR related cost — Main result | AMR related cost — Sensitivity Analysis | Proxy outcomes | Study limitations |
|---|---|---|---|---|---|---|---|---|---|---|---|
| CDC [36] | NR | NR | 10 years | NR | Cephalosporin-resistant NG | Unclear | NG cephalosporin resistance prevalence becomes widespread | Direct medical costs over 10-year period associated with widespread cephalosporin-resistant NG is estimated to be at least $235 million | NR | NR | NR |
| Turner et al. [11] | •Current management: patients treated with dual Ceftriaxone/azithromycin treatment either due to symptoms and positive microscopy or as contacts of infected individuals; some patients are treated on the same day, others wait for lab results. •Simple POCT management: all patients tested and treated on same day as though resistant to current management. •AMR POCT management: all patients tested with AMR POCT to identify NG infections that do not require treatment with ceftriaxone. | Published literature, GRASP and PHE data, and data from clinical practice. (GBP 2014) | NR | Healthcare provider | NG ciprofloxacin or penicillin resistance | Main analysis —cost of AMR POCT | •All NG infections are fully (100%) treated with ceftriaxone and azithromycin due to >5% resistance in alternative regimens (such as ciprofloxacin). •Prevalence of NG resistance to ciprofloxacin (37%) and penicillin (23%). •AMR POCT for assessing resistance to either ciprofloxacin or penicillin incurs an additional £25 testing cost. | Total cost: •Current management— £195,969,677 •AMR POCT management— £230,406,720 •Increased cost with AMR POCT— £34,437,043 | NR | Reduction in number of ceftriaxone treatments annually, n (%): •AMR POCT ciprofloxacin: 22,054 (66%) •AMR POCT penicillin: 26,499 (79%) | •Static and simplified model. •The indirect effects of reduced NG transmission to partners or on NG complications were not considered. •Only AMR POCT which integrates with the existing POCT technology was considered. •The longer-term effects of changing treatment strategy on the evolution of drug resistance over time were not considered. |

(*Continued*)

**Table 7.** (Continued)

| Author | Strategies/ interventions/ comparators | Source of Cost | Horizon (Discount rate) | Perspective | Definition of AMR | Method of AMR incorporation | AMR Assumption | AMR related cost | | Proxy outcomes | Study limitations |
|---|---|---|---|---|---|---|---|---|---|---|---|
| | | | | | | | | Main result | Sensitivity Analysis | | |
| Allan-Blitz et al. [12] | •*gyrA* assay + targeted therapy •Standard two-drug therapy | Interview of clinic and laboratory personnel, and published literature | NR | Healthcare provider | NG ciprofloxacin-resistance: Mutation on codon 91 gyrase A gene (*gyrA*) detected by *gyrA* assay. | Main analysis —cost of *gyrA* assay | Indeterminate or *gyrA* mutant genotypes were treated with ceftriaxone and azithromycin. | Per-case cost: •*gyrA* assay + targeted therapy: $155.16 •Standard two-drug therapy: $197.19 | NR | NR | •Did not account for all inherent costs of screening and treatment. •Results applicable only to settings with established testing programs. •Small sample size (n = 234 unique cases). |
| Chesson et al. [38] | •Scenario 1: 2% annual ceftriaxone resistance for 10 years. •Scenario 2 "emerging resistance": ceftriaxone resistance increases to 15% over 10 years. | Published literature. (USD 2016) | 10 years (3% annually) | NR | Ceftriaxone resistance | | •820,000 cases of gonorrhoea annually in the absence of resistance. •Scenario 1: 2% prevalence of resistance is maintained over 10 years. •Scenario 2: linear increase in prevalence of ceftriaxone resistance from 2% in year 0 to 15% in year 6, remaining constant at 15% through to year 10. •Emerging ceftriaxone resistance was assumed to be similar to ciprofloxacin resistance of the 1990 and 2000s. | Total additional cost of emerging resistance, scenario (2 vs 1) —($ million): •Year 1: $4.9 •Year 2: $12.4 •Year 3: $21.1 •Year 4: $30.4 •Year 5: $39.9 •Year 6: $49.4 •Year 7: $53.9 •Year 8: $55.5 •Year 9: $55.6 •Year 10:$54.9 •Total over 10 years: $378.2 | •Varying base case NG incidence (395,000 to 1,245,000) per annum resulted in total 10-year additional costs of $182.2 to $574.2 million. •Varying NG peak resistance from 15% (base case) to between 5% and 20% resulted in total 10-year additional costs of $81.9 to $540.8 Million. •Multiway SA; total 10-year additional costs of $41 to $1099 Million | NR | •Simple and static model. •Emerging cephalosporin resistance was modelled based on ciprofloxacin resistance between the 1990s and 2000s. •Model assumption for gonorrhoea-attributable HIV was simple and dated. •The costs to develop, implement and maintain programs that sustain NG ceftriaxone resistance at less than 2% were not included |

*(Continued)*

**Table 7.** (Continued)

| Author | Strategies/ interventions/ comparators | Source of Cost | Horizon (Discount rate) | Perspective | Definition of AMR | Method of AMR incorporation | AMR Assumption | AMR related cost | | Proxy outcomes | Study limitations |
|---|---|---|---|---|---|---|---|---|---|---|---|
| | | | | | | | | Main result | Sensitivity Analysis | | |
| Wynn and Klausner [40] | •gyrA assay + targeted therapy •Standard of care—NG positive patients receive dual therapy | Centers for Medicare & Medicaid fee schedule; National Average Drug Acquisition Cost data set; and Injection procedure and follow-up visit. | NR (no discounting) | Healthcare provider | Wild-type gyrA predicted NG susceptibility to ciprofloxacin. Non- wild-type gyrA predicted resistance to ciprofloxacin. | Main analysis —cost of gyrA assay. SA- varying NG prevalence for low/high risk groups; and including a scenario where 30% gyrA test results were indeterminate. | •2% NG test positivity •Prevalence of NG Ciprofloxacin susceptibility (70%) •non–wild-type gyrA received dual therapy ceftriaxone and azithromycin | •The average cost of dual therapy (Standard care) per patient was $72. •The cost at which incorporating gyrA testing would breakeven was $50 per test. gyrA assay less than $50 would be cost saving. •If 30% of tests were indeterminant, the breakeven cost was $35/ test. | •Prevalence of ciprofloxacin susceptibility at 55%, 85% and 95% resulted in breakeven cost of $39, $60, and $67 respectively. •If 30% of tests were indeterminant, the breakeven cost was $35/ test for the base case and ranged from $8 to $79 in other scenarios | NR | •Cost estimates were limited to those provided by centres of Medicare and Medicaid fee. •The model algorithm used assumes that patients prescribed ciprofloxacin do not return for follow-up compared to dual-therapy where a second visit is required. •The study did not consider differential treatment uptake and time to treatment between the 2 arms. |

AMR, antimicrobial resistance; CDC, Centers for Disease Control and Prevention; GRASP, the gonococcal resistance to antimicrobials surveillance programme; NA, not applicable; NG, gonorrhoea; NR, not reported; PHE, Public Health England; POCT, point-of-care test; SA, sensitivity analysis; SC, standard of care; SHC, sexual health clinic; STD, sexually transmitted disease; TV, Trichomoniasis; US, United States; USD, United States dollar.

**Table 8. Methodological specifications of economic evaluations and cost effectiveness of AMR in the treatment of curable STIs (by year).**

| Author | Strategies/interventions/comparators | Model type | Horizon (Discount rate) | Perspective | Definition of AMR | Method of AMR incorporation | AMR Assumptions | AMR related findings | | Study Limitations |
|---|---|---|---|---|---|---|---|---|---|---|
| | | | | | | | | Base case | Sensitivity Analysis (SA) | |
| Phillips et al. [31] | •Endo-cervical culture for NG. •No-test strategy. | Decision tree | 5 years (5% annually) | NR | NG Penicillin resistance | SA—more expensive antibiotics would be required to treat penicillin-resistant organisms. Therefore, the costs associated with treating patients with positive test results for gonococcal infection were increased | More expensive antibiotics would be required to treat penicillin-resistant organisms Culture test would include antibiotic sensitivity testing. Patients with antibiotic resistance would be treated appropriately | Threshold analysis—Testing women for gonorrhoea would reduce overall costs if the prevalence of infection exceeded 1.5%. Direct medical cost would be reduced by routine testing if infection exceeded 2.5%. | Varying the cost of initial treatment from $100 to $140 (base case, $120) for patients with positive test results had little effect on the threshold prevalence, which ranged from 1.4%-1.6%. | NR |
| Nettleman et al. [32] | •Strategy 1: β-lactamase screening + ciprofloxacin treatment for PPNG and ampicillin/probenecid for other NG infections •Strategy 2: strategy 1, but ceftriaxone substituted for ciprofloxacin •Strategy 3: no β-lactamase screening, ampicillin/probenecid only treatment •Strategy 4: no β-lactamase screening + ciprofloxacin only treatment •Strategy 5: no β-lactamase screening + ceftriaxone only treatment | Decision analysis | NR | NR | Penicillinase producing NG—PPNG | •Main analysis—Cost effectiveness of beta-lactamase screening. •SA—varied the prevalence of PPNG isolates. | •5% of NG isolates were assumed to be penicillin resistant. •Cost of β-lactamase screening was $0.50/test. •Ceftriaxone and ciprofloxacin were effective against penicillin-resistant NG strains. •Ampicillin was assumed to be completely ineffective against penicillin-resistance. •Ampicillin, ceftriaxone and ciprofloxacin were completely effective against penicillin-susceptible NG strains. | Cost-effectiveness values‡: •Strategy 1: $2.26 •Strategy 2: $2.31 •Strategy 3: $4.91 •Strategy 4: $2.11 •Strategy 5: $3.65 | •The use of ceftriaxone became more cost-effective compared to strategies involving β-lactamase screening if the probability of β-lactamase production by NG gonorrhoeae isolates exceeded 21%, the cost of β-lactamase screening exceeded $5.80 •Use of ciprofloxacin as the sole anti-gonococcal agent remained more cost-effective than strategies involving β-lactamase screening until the proportion of isolates producing penicillinase was <3%. •Ampicillin/probenecid as the sole anti-gonococcal therapy without screening for β-lactamase was most cost-effective if <0.25% of isolates produced penicillinase. | NR |
| Crabbe et al. [33] | •Gold standard: all male patients are treated with cefixime and doxycycline for GU and NGU respectively •Specificity: male patients are treated with doxycycline for NGU or cefixime for GU based on urethral smear examination •Financial accessibility: all male patients are treated with cotrimoxazole or kanamycin for GU and doxycycline for NGU. Cefixime is reserved as 2nd line treatment for non-response to first GU treatment. | Decision tree | NR | NR | NR | SA—AMR was considered to vary widely across countries or types of setting, therefore the impact of varying treatment efficacy values for cotrimoxazole and kanamycin were assessed on the 'Financial accessibility' strategy | NG resistance to antibiotics: •Baseline: •Cefixime: 0% •Cotrimoxazole: 30% •Kanamycin: 30%; •SA: •Cotrimoxazole: 0%, 10% and 20% •Kanamycin: 0%, 10% and 20% | Cost per cured urethritis: •Gold standard: $6.8 •Specificity: $5.8 •Financial accessibility: $3.3 | Impact of cotrimoxazole/kanamycin efficacy on financial accessibility strategy; cost per cured urethritis: •70% efficacy (baseline): $3.3/$3.2 •80% efficacy: $3.0/$3.0 •90% efficacy: $2.8/$2.7 •100% efficacy: $2.5/$2.4 | •Simplified model. •Infection transmission to partners and the rate of complications were not considered. •Extra costs associated with improving the follow-up visit rate or with monitoring gonococcal antimicrobial resistance were not considered. •Low labour costs are representative of many low-income countries and not generalisable. •Intramuscular administration vs. oral intake was not considered. |

(*Continued*)

**Table 8.** (Continued)

| Author | Strategies/interventions/comparators | Model type | Horizon (Discount rate) | Perspective | Definition of AMR | Method of AMR incorporation | AMR Assumptions | AMR related findings — Base case | AMR related findings — Sensitivity Analysis (SA) | Study Limitations |
|---|---|---|---|---|---|---|---|---|---|---|
| Roy et al. [34] | •Strategy 1: ciprofloxacin + culture tests + ciprofloxacin susceptibility tests •Strategy 2: ciprofloxacin + non-culture tests •Strategy 3: ceftriaxone + culture tests + ceftriaxone susceptibility tests •Strategy 4: ceftriaxone + non-culture tests | Decision tree | NR | Healthcare system | NG ciprofloxacin resistance | Main analysis—cost of AMR susceptibility test; SA—varied the prevalence of ciprofloxacin-resistant NG | •Base case prevalence of ciprofloxacin-resistant NG (0.1%) •Base case prevalence of ceftriaxone-resistant NG (0%) •80% of culture-positive specimens would also be tested for AMR susceptibility when ciprofloxacin was used for treatment. •Symptomatic women infected with a resistant strain recalled and retreated for strategies 1 and 3 •Infection with a resistant strain led to complete treatment failure | ICER/additional case of PID prevented vs baseline: •NG prevalence (0.01) •Strategy 1: baseline •Strategy 2: \$73,478 •Strategy 3: strongly dominated •Strategy 4: \$8,070,000 •NG prevalence (0.10) •Strategy 1: strongly dominated •Strategy 2: baseline •Strategy 3: strongly dominated •Strategy 4: \$173,000 | The findings were sensitive to the cost of antimicrobial agents, diagnostic tests, gonorrhoea prevalence and ciprofloxacin-resistance prevalence. •When there were greater combinations of gonorrhoea prevalence and ciprofloxacin resistance prevalence than the base case and the ratio of ceftriaxone to ciprofloxacin cost reduces, non-culture strategies (2&4) became optimal (cost per successfully treated patient). •When non-culture tests cost 3 times that of culture test, then culture tests (including susceptibility testing)—strategies 1&3 became optimal for all combination of gonorrhoea prevalence and ciprofloxacin-resistance prevalence. •Culture based strategies (1&3) were optimal if prevalence of gonorrhoea was <6% regardless of the differences in the diagnostic performance of culture and non-culture based tests. •When the prevalence of gonorrhoea approaches 3%, strategy 1 is the optimal strategy if the prevalence of resistance is <4%. If ciprofloxacin-resistance levels are ≥3% and gonorrhoea prevalence is >13%, a switch to strategy 4 is recommended. | The costs and benefits associated with diagnosis and treatment of both NG and CT were not considered. Analysis was limited to adult women. Monte Carlo simulations on AMR drug selection were based on assumed distributions and not actual data. Ciprofloxacin resistance was assumed to result in 100% treatment failure, and that ceftriaxone resistance was zero The model assumes that when ceftriaxone-resistant gonorrhoea becomes problematic, an equally effective and affordable antimicrobial agent will be available to replace ceftriaxone. |
| Price et al [35] | Metronidazole treatment for •Group A—men with urethritis •Group B—men with urethritis and genital ulcer disease •All—all male attendees of the STD clinic •Standard care—asymptomatic male partners of symptomatic women treated with metronidazole. | Decision tree | NR | Govt. | Potential TV metronidazole resistance | SA—Efficacy of metronidazole was varied from 90%-100% due to some published evidence on TV metronidazole resistance | Base case clinical efficacy of metronidazole was assumed to be 100% | ICER/HIV case averted, metronidazole vs standard care: •Group A: \$32.27 •Group B: \$15.85 •All: \$15.43 | ICER/HIV case averted, metronidazole vs standard care:95% efficacy: •Group A: \$32.23 •Group B: \$16.76 •All: \$16.35 90% efficacy: •Group A: \$34.29 •Group B: \$17.80 •All: \$17.41 | •HIV transmission beyond the primary HIV infection attributable to TV was not considered. •CEA does not consider TV transmission or the effect of untreated TV infections •Cost limited to payer (Govt.) and future cost related to HIV morbidity were not considered. •Data source limited to data from a single STD clinic site |
| Xiridou et al. [37] | Spread of AMR to first-line treatment and QALYs for •Ceftriaxone and azithromycin •Ceftriaxone monotherapy | Transmission and economic model | 10–60 year horizon (costs were discounted by 4% and QALY by 1.5% annually) | Healthcare provider | Clinical resistance: Individuals infected with a strain resistant to an antibiotic cannot be cured if treated with that antibiotic | Main analysis—a scenario (baseline) with no initial azithromycin resistance. SA—additional scenario with 5% initial azithromycin resistance. | For the baseline scenario, dual therapy is introduced in the absence of resistance to ceftriaxone and azithromycin In the first 8 years that ceftriaxone was the first-line treatment for NG, there were no cases with ceftriaxone resistance. Probability of resistance to one antibiotic: $10^{-10} - 10^{-6}$ Probability of resistance to two antibiotics: $(10^{-10} - 10^{-6})^2$ | Cumulative ICER/QALY by time horizon, dual vs monotherapy: •10 years: €9.74 x $10^8$ •20 years: €6,413,679 •30 years: €151,990 •40 years: €79,705 •50 years: €41,421 •60 years: €14,866 | Scenario analysis, 5% initial azithromycin resistanceCumulative ICER/QALY by time horizon, dual vs monotherapy: •10 years: €1.09 x $10^9$ •20 years: €5,499,464 •30 years: €1,890,572 •40 years: €1,393,843 •50 years: €1,286,210 •60 years: €1,224,768 | •Different anatomic locations of NG infections were not considered. •Assortative partner selection was not considered. •The risk of imported NG infections with resistant strains from outside the Netherland was not considered. •The development of resistance was limited to ceftriaxone and azithromycin. |

(*Continued*)

**Table 8.** (Continued)

| Author | Strategies/interventions/comparators | Model type | Horizon (Discount rate) | Perspective | Definition of AMR | Method of AMR incorporation | AMR Assumptions | AMR related findings | | Study Limitations |
|---|---|---|---|---|---|---|---|---|---|---|
| | | | | | | | | Base case | Sensitivity Analysis (SA) | |
| Harding-Esch et al. [39] | •Standard of care (SC)—IM ceftriaxone and oral azithromycin<br>•Dual therapy, including ceftriaxone:<br>•Strategy A. AMR POCT for ciprofloxacin resistance only—infections resistant to ciprofloxacin receive SC, non-resistant infections receive ciprofloxacin and ceftriaxone.<br>•Strategy B Dual AMR POCT for azithromycin and ciprofloxacin resistance—non azithromycin resistance receives SC, if azithromycin resistant, ciprofloxacin and ceftriaxone is given. If azithromycin and ciprofloxacin resistant, ceftriaxone only treatment is given.<br>•Strategy C Dual AMR POCT for ciprofloxacin and azithromycin resistance—no ciprofloxacin resistance, ciprofloxacin and ceftriaxone is given. If ciprofloxacin resistant, SC is given. Resistant to ciprofloxacin and azithromycin, ceftriaxone alone is given.<br>•Monotherapy optimisation<br>•Strategy D: AMR POCT for azithromycin resistance—no resistance, azithromycin is given. If azithromycin resistant, ceftriaxone and ciprofloxacin is given. False negative AMR test would receive ceftriaxone at test of cure<br>•Strategy E: AMR POCT for ciprofloxacin—if no resistance, ciprofloxacin only is given. If ciprofloxacin resistant, SC is given. False negative AMR test would receive ceftriaxone at test of cure. | Decision Tree | Initial patient treatment | Healthcare provider | Assay for molecular determinants of NG AMR:<br>•gyrA for ciprofloxacin resistance<br>•23S rRNA and mtrCDE transporter for azithromycin resistance.<br>Absence of these determinants predicts susceptibility to ciprofloxacin and azithromycin respectively. | Main analysis—Cost for AMR POCTs<br>SA—varying the cost of POCTs and the prevalence of azithromycin and ciprofloxacin resistance. | There is no ceftriaxone resistance and ceftriaxone is an effective (100%) cure for NG. | Additional cost per optimal treatment gained compared to standard care<br>•Strategy A: £1,226.97 (€1,640.81)<br>•Strategy B: £745.44 (€996.87)<br>•Strategy C: £835.39 (€1,117.16)<br>•Strategy D: £414.67 (€554.53)<br>•Strategy E: £671.82 (€898.42)<br>Additional cost per ceftriaxone-sparing treatment compared to standard care<br>•Strategy A- dominated<br>•Strategy B: dominated<br>•Strategy C: dominated<br>•Strategy D: £11.29 (€15.09)<br>•Strategy E: £22.94 (€30.68) | The greatest impact on cost-effectiveness was;<br>•The prevalence of azithromycin resistance, >3% resistance increased the ICER.<br>•AMR POCT sensitivity at 90% increased the cost per optimal treatment exponentially.<br>•If the prevalence of ciprofloxacin resistance increased by 20%, there was an exponential increase in cost per optimal treatment for using strategies A and E in women.<br>•The cost of single detection AMR POCT—monotherapy strategies became cost saving when AMR POCT reduced from base case value of £29 to £18 and £16 for strategies D and E respectively. | •NG AMR data were limited to previously published data.<br>•AMR POCTs were still in development phase. Some of the model's epidemiological parameters are likely to change by the time the AMR POCTs are available for use in routine practice.<br>•Data limited to England.<br>•Limited to cost associated with uncomplicated GC infection.<br>•Patients limited to GC only infection and co-infection with other STIs were not considered.<br>•Time horizon limited to patient treatment only.<br>•Costs incurred outside of the SHC, and costs associated with changing clinical pathways in order to accommodate the AMR POCTs were not accounted for. |

AMR, antimicrobial resistance; CEA, cost-effectiveness analysis; CT, chlamydia; HIV, human immunodeficiency virus; ICER, incremental cost-effectiveness ratio; IM, intramuscular injection; NA, not applicable; NG, gonorrhoea; NGU, non-gonococcal urethritis; NR, not reported; PID, pelvic inflammatory disease; PPNG, penicillinase-producing Neisseria gonorrhoeae; POCT, point-of-care test; QALY, quality adjusted life year; SA, sensitivity analysis; SC, standard of care; SHC, sexual health clinic; STD, sexually transmitted disease; TV, Trichomoniasis; US, United States; USD, United States dollar.

¥ Calculated as total cost of strategy/utility value (score of zero assumed for uncured gonococcal infection; score of 1 for persons cured or never infected)

i. *Definition of gonorrhoea AMR*: AMR was defined in broad terms in some studies and more specifically in others (Tables 7 & 8). Broad definitions included gonorrhoea cephalosporin resistance [36], ceftriaxone resistance [38], ciprofloxacin resistance or penicillin resistance [11], and gonorrhoea treatment failure [35, 37]. When AMR was defined in this manner it was often unclear what specific method (treatment failure or antibiotic susceptibility testing with minimum inhibitory concentration breakpoints) was used to establish resistance because AMR was frequently linked to published epidemiological estimates from sentinel surveillance programmes without providing additional detail. Other studies used gonorrhoea resistance specific genetic/molecular phenotypes such as mutant gyrA for ciprofloxacin resistance [12, 39, 40], 23SrRNA and mtrCDE for azithromycin resistance [39] or culture for penicillinase producing *Neisseria gonorrhoeae* [31] to define resistance.

ii. *Method of gonorrhoea AMR incorporation into the economic evaluation*: The most common approach for incorporating AMR was to account for the additional cost of AMR testing and/or treatment [11, 12, 37, 39]. Threshold analyses were also reported to identify the breakeven cost of using AMR POCT [40] and the threshold prevalence of gonorrhoea for using culture based testing [31]. Mostly, a baseline prevalence of resistance to certain drugs in a base case scenario was incorporated in the main model, and then altered in a sensitivity analysis [32–34, 38, 40]. Others made assumptions around the baseline AMR test accuracy [12, 39] and cost per test [12, 32], and varied these in the sensitivity analysis [32, 39]. One study utilised a more complex methodology by creating two scenarios—one with and the other without baseline azithromycin resistance, and developed a gonorrhoea transmission model which was then used to inform the economic model [37].

iii. *Gonorrhoea AMR assumptions*: Variable but inconsistent assumptions about resistance were made, including the prevalence of gonorrhoea AMR (in 9/13 studies [11, 32–34, 36–38, 40, 42]), what therapy would be used when AMR was present (in 4/12 studies [12, 32, 34, 37]), and the degree of clinical effectiveness for modelled antibiotics (in 2/12 studies [35, 39]).

iv. *Study reported limitations*: Of the reported study limitations, the transferability of epidemiological data on AMR prevalence in one area to other geographical locations, and uncertainty in predicting future AMR prevalence were acknowledged [38, 39]. The use of static economic models which do not take into account the complexities of STI transmission dynamics were common [11, 33, 38]. The limited settings in which AMR POCT can be utilised due to cost and required technical expertise were acknowledged [39] and the lack of data on how introducing new treatment strategies might affect the evolution of gonorrhoea AMR was also identified [11].

v. *Study perspectives and horizons*: Of the eight studies which reported a perspective, all took the perspective of the provider (healthcare system, sector or government). The study horizons ranged from the time of treatment (most commonly) to a 60 year time period (Tables 7 & 8).

**Quality assessment.** The reporting quality of both the cost and economic evaluation studies varied. For cost studies, a sensitivity analysis was included in 2/5 studies, and the descriptions of the data sources, analytical methods, valuation techniques utilised and the method used to estimate the costs associated with AMR was limited in one of these [36] (S3 File). For the economic evaluations, the study population, choice of model and assumptions were appropriately reported, however information on study perspective, horizon, discount rate and study limitations was often limited (S3 File).

**Table 9. Summary of results of base case analysis and sensitivity analyses.**

| Trial Arm | Average cost per patient (£) | % cleared of infection at 2 weeks | ICER |
|---|---|---|---|
| Ceftriaxone | 9.41(2.00–24.16) | 98 | Dominates |
| Gentamicin | 13.25(2.44–36.94) | 91 | |

## Case study accounting for AMR cost in the treatment of gonorrhoea

**Initial analysis without accounting for AMR.** In the GToG trial a higher proportion of patients treated with ceftriaxone compared with gentamicin had microbiological cure at the 2-week follow up (98% vs 91%). The average cost per patient treated with gentamicin was £13.25, compared with £9.41 for those treated with ceftriaxone. The higher cost of gentamicin treatment was due to the cost of additional consultations and treatment of patients with persistent infection at the 2-week follow up. Treatment with gentamicin was therefore not non-inferior to ceftriaxone and it was not cost-neutral (Table 9). Fig 4 shows the results of the probabilistic sensitivity analysis involving 1000 simulations. Most of the points were in the top left-hand quadrant, indicating that treatment with ceftriaxone dominated treatment with gentamicin confirming that gentamicin is not shown to be non-inferior and is unlikely to be cost-neutral.

**Analysis accounting for AMR.** On Table 10, we show the results when accounting for AMR in the economic evaluation of an alternative treatment such as gentamicin compared to ceftriaxone, as the current standard of care, for the treatment of gonorrhoea.

i.  The potential additional cost of treating a patient with ceftriaxone resistant gonorrhoea was estimated to be £190.54 (Table 5). This cost was applied to the trial data, assuming that all those who experienced a treatment failure in the ceftriaxone arm had a resistant strain of gonorrhoea. As expected, including the potential costs of resistance, for those who were not successfully treated during their initial treatment, increased the overall costs per patient treated in this arm (Table 10). However, as we assumed that only those who had an initial treatment failure would experience the additional costs associated with resistance, ceftriaxone remained the cheaper treatment.

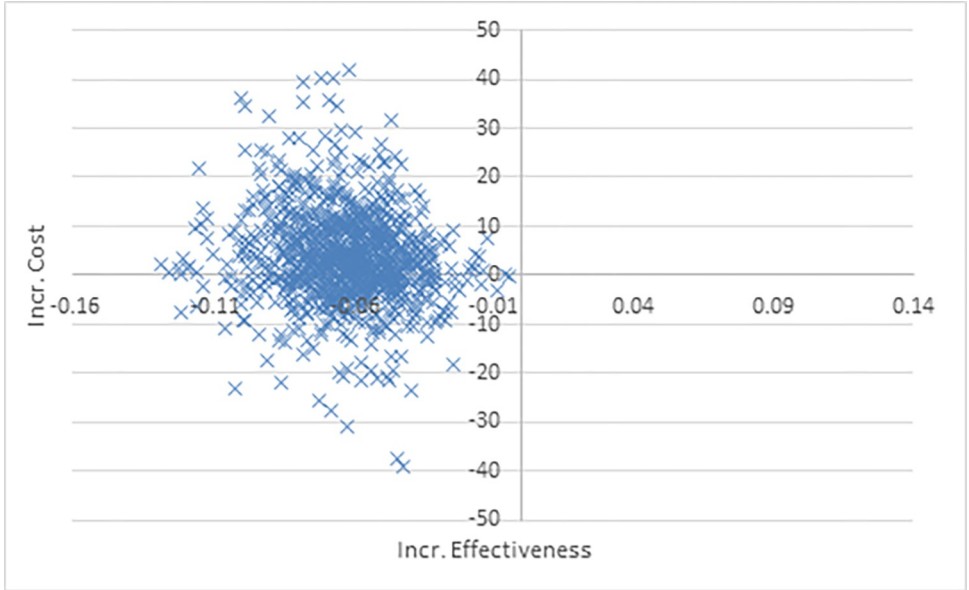

**Fig 4. Incremental cost-effectiveness scatterplot for clearance of infection–gentamicin vs. ceftriaxone.**

**Table 10. Analysis accounting for AMR.**

| | Original value | Revised value | Ceftriaxone arm: Average cost per patient | Gentamicin arm: Average cost per patient |
|---|---|---|---|---|
| Base case | - | - | £9.41 | £13.25 |
| a) Including costs for treating resistant infections for those without clearance of *N. gonorrhoeae* in the ceftriaxone arm | £9494.10 | £164.20 | £13.17 | |
| b) Varying the rates of clearance of *N. gonorrhoeae* | Gent. Arm: 91% Cef. Arm: 98% | Gent. Arm: 72%-97% Cef. Arm: 92%-99% | £14.00-£7.79 | £29.37-£8.72 |
| c) Increasing the cost of ceftriaxone treatment (including an additional penalty to protect this class of antibiotics) | | £.80-£20.53 | £10.21-£29.94 | |

Costs are £UK (2020/21). Gent. = gentamicin, Cef. = ceftriaxone.

ii. In a threshold analysis we varied the rates of clearance of infection for ceftriaxone. This demonstrated that clearance rates for gentamicin would need to be higher than those for ceftriaxone for the treatment to be cost-neutral, due to the higher initial treatment costs associated with gentamicin treatment. If the clearance rate for ceftriaxone dropped to 92% then gentamicin became a cost-neutral treatment.

iii. To assess the societal cost of resistance we applied the range of costs estimated for ceftriaxone resistance from a previous study [18] to those in the ceftriaxone arm of the trial which resulted in the average cost per patient ranging from £10.21 to £29.94. Our analysis determined that the cost of ceftriaxone would need to increase to £8.44 in order for treatment with gentamicin to be cost neutral.

## Discussion

We conducted a systematic review of the economic evidence for measuring AMR in curable STIs and developed a preliminary economic evaluation model which incorporates gonorrhoea AMR to assess an alternative antimicrobial regimen for gonorrhoea treatment. We found a small number of studies were eligible for inclusion, suggesting that the existing evidence base is limited. The majority of the included studies (11/12) related to gonorrhoea, possibly reflecting the high prevalence and clinical importance of gonorrhoea resistance [4].

Cost estimates from the United States indicate that the cost of gonorrhoea resistance to current first-line treatment (ceftriaxone) is substantial, equating to $235 to $378.2 million in direct medical costs over 10 years [36, 38]. These estimates are likely to be significantly higher in low and middle income settings [18], and if indirect costs and the societal implications of AMR were included. Gonorrhoea resistance testing and improved antibiotic stewardship were frequently reported as strategies to extend the time period over which ceftriaxone remains effective by restricting its widespread use. A 70% reduction in ceftriaxone use could potentially be achieved using molecular based AMR point-of-care testing to guide the use of quinolone therapy but this approach was not found to be cost-effective when compared to ceftriaxone/azithromycin dual therapy [11, 12, 39, 40]. Cost-effectiveness was reduced further if there was an increase in the prevalence of quinolone resistance or cost of point-of-care tests, or if AMR test performance was lower than predicted. However, these analyses were restricted to evaluating

direct medical costs, had a short time horizon, used static models, and did not include the broader societal impact of AMR which limits their interpretation.

We found the methodology used in existing cost/economic evaluations varied considerably, including in terms of how AMR was defined, assumed projections of AMR prevalence, costs included, and the perspective adopted. How AMR is defined presents a specific challenge since molecular, microbiological and clinical definitions of resistance can vary significantly and with no single 'gold standard' measure [43]. The modelling assumptions around AMR in gonorrhoea were mainly related to the prevalence of resistance, cost of treatment and proposed treatment pathway when resistance occurred. AMR prevalence estimates were obtained from a variety of sources which could have been influenced by laboratory methodology, frequency of testing and/or the underlying healthcare delivery system [44]. Cost estimates were sensitive to changes in AMR prevalence in most studies highlighting the importance of surveillance systems which are representative of the general population and utilise a robust methodology. The cost of treatment was sourced from either primary and/or secondary sources, but the assumptions around patient management pathways when AMR was suspected varied significantly with no commonly accepted 'best' approach for managing patients who had failed first line therapy. The most frequent perspective adopted was that of the healthcare provider, but this may underestimate the total cost of resistance [16] by not including the full cost of morbidity, loss of income, reduced productivity and use of antibiotic prophylaxis. A more consistent and comprehensive approach to incorporate patient and societal costs of AMR is therefore desirable [14, 16, 42].

Our case study explored different approaches to incorporate AMR using data from a recent large multicentre RCT in which patients were treated with ceftriaxone or gentamicin [24]. We conducted a threshold analysis and found gentamicin treatment to be cost neutral if the failure rate for ceftriaxone increased to 10% (from 2%) which is consistent with reports by Wynn and Klausner [40] who found the cost neutrality for ciprofloxacin resistance genetic testing was dependent on the prevalence of ciprofloxacin resistance but not gonorrhoea prevalence. A limitation of most previous studies was the lack of inclusion of societal costs of AMR. Given that ceftriaxone is the only remaining reliably effective therapy for gonorrhoea, and that continued use increases the risk of subsequent ceftriaxone AMR development, we assessed the societal cost—i.e. direct and indirect costs, resistance modulating factors and rate of consumption of antibiotics that drive resistance [18]. We identified that if the cost of ceftriaxone was increased to £8.44 from £4.60, treatment with gentamicin would become cost-neutral. By accounting for gonorrhoea AMR via multiple approaches and using prospective data, it is therefore possible to provide additional useful information for decision-makers about when alternative antibiotics might be considered as a replacement for standard treatment.

We recognise a number of potential limitations. The systematic review was restricted to studies published in English, and the search terms chosen were based on scoping searches and prior knowledge of the literature. The use of additional search terms may have increased the number of records returned but would have made the number of records requiring review unfeasible and was considered unlikely to identify other highly relevant studies. For the economic evaluation of gentamicin as an alternative to ceftriaxone, a static rather than dynamic model structure was adopted. Although a dynamic model would have allowed the wider impacts of resistance to be considered, it was beyond the scope of this exploratory study which aimed to assess different approaches for incorporating the effects of AMR. A further limitation was the short follow up period associated with the clinical trial, which did not allow data to be gathered on the longer-term management of patients who had failed treatment. We addressed this by assuming that management would follow current clinical management pathways. Also, as our case study was conducted in a high income setting, the findings may not be

generalizable to low income settings but contributes to the debate on how future economic evaluations could more fully incorporate the economic impact of AMR for curable STIs.

There is a dearth of data in relation to the economic analysis of AMR for curable STIs. However, by incorporating AMR costs (including in our exemplar study on alternative treatment for gonorrhoea) more robust interpretations in relation to the costs/cost effectiveness of new or alternative treatment strategies for curable STIs can be made. The current evidence relating to the economic impact of AMR for curable STIs is generally limited to direct, aggregated costs over a short period, obtained from high income countries, is largely specific to gonorrhoea AMR, and often lacks a societal perspective. There is no standardised approach for the measurement of AMR in patients with curable STIs which results in uncertainty when reporting on the economic impact of AMR and makes comparisons between studies difficult. Further research is required to inform guidance on optimal approaches to capture AMR costs for curable STIs (e.g. how AMR is defined and how AMR associated costs are adequately captured) and methodically incorporate such costs into economic evaluations.

## Supporting information

**S1 File. Search strategy.**
(DOCX)

**S2 File. Study selection.**
(DOCX)

**S3 File. Quality assessment of included studies.**
(DOCX)

**S4 File. Assumptions for the economic evaluation comparing gentamicin to ceftriaxone for the treatment of gonorrhoea.**
(DOCX)

**S1 Table. Data extraction form.**
(DOCX)

## Acknowledgments

The authors thank the whole GToG team, Pollyanna Ford, as her MSc dissertation helped lay the foundations for aspects of this paper, and Lena Schnitzler for providing the English translation for the Müller et al. [23] checklist for cost-of-illness.

## Author Contributions

**Conceptualization:** Jonathan D. C. Ross, Louise Jackson.

**Data curation:** Oluseyi Ayinde, Louise Jackson.

**Formal analysis:** Oluseyi Ayinde, Louise Jackson.

**Investigation:** Oluseyi Ayinde.

**Methodology:** Oluseyi Ayinde, Jonathan D. C. Ross, Louise Jackson.

**Writing – original draft:** Oluseyi Ayinde.

**Writing – review & editing:** Oluseyi Ayinde, Jonathan D. C. Ross, Louise Jackson.

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
