## [Decision Letter · Decision Letter 0]

6 Jun 2023

PONE-D-23-04588Economic evaluation of antimicrobial resistance in curable sexually transmitted infections; a systematic review and a case studyPLOS ONE

Dear Dr. Ayinde,

Thank you for submitting your manuscript to PLOS ONE. After careful consideration, we feel that it has merit but does not fully meet PLOS ONE’s publication criteria as it currently stands. Therefore, we invite you to submit a revised version of the manuscript that addresses the points raised during the review process.

We look forward to receiving your revised manuscript.

Kind regards,

Kwame Kumi Asare, Ph.D

Academic Editor

PLOS ONE

Journal Requirements:

"The original RCT was funded by the National Institute for Health Research Health Technology Assessment Programme (project number 12/127/10). The views and opinions expressed therein are those of the authors and do not necessarily reflect those of the Health Technology Assessment Programme, NIHR, NHS or the Department of Health.."

"JDCR reports personal fees from GSK Pharma and Hologic Diagnostics, as well as ownership of shares in GSK Pharma and Astrazeneca Pharma; and is author of the UK and European Guidelines on Pelvic Inflammatory Disease; is a Member of the European Sexually Transmitted Infections Guidelines Editorial Board. He is an NIHR Journals Editor and associate editor of Sexually Transmitted Infections journal. He is an officer of the International Union against Sexually Transmitted Infections (treasurer). LJ and OA report no conflicts of interest."  

We note that you received funding from a commercial source: GSK Pharma, Hologic Diagnostics, Astrazeneca Pharma

Within this Competing Interests Statement, please confirm that this does not alter your adherence to all PLOS ONE policies on sharing data and materials by including the following statement: "This does not alter our adherence to PLOS ONE policies on sharing data and materials.” (as detailed online in our guide for authors http://journals.plos.org/plosone/s/competing-interests).  

If there are restrictions on sharing of data and/or materials, please state these. Please note that we cannot proceed with consideration of your article until this information has been declared. 

6. Please remove your figures from within your manuscript file, leaving only the individual TIFF/EPS image files, uploaded separately. These will be automatically included in the reviewers’ PDF. 

Reviewers' comments:

Reviewer's Responses to Questions

**Comments to the Author**

1. Is the manuscript technically sound, and do the data support the conclusions?

Reviewer #1: Yes

Reviewer #2: Yes

2. Has the statistical analysis been performed appropriately and rigorously? 

Reviewer #1: Yes

Reviewer #2: Yes

3. Have the authors made all data underlying the findings in their manuscript fully available?

Reviewer #1: Yes

Reviewer #2: Yes

4. Is the manuscript presented in an intelligible fashion and written in standard English?

Reviewer #1: Yes

Reviewer #2: Yes

5. Review Comments to the Author

Reviewer #1: Congratulations for selecting a research topic of great interest with results that may have potential implications for clinical practice and public health policies.

The systematic review methodology follows current evidence based practice, is well described and comprehensively documented in tables and supplementary files.

For PLOS ONE readers is highly relevant to be aware that there are no consistent methodological approaches in various economic analyses which could lead to variable results and thus recommendations. It is instructive to learn about the parameters influencing the results which are well described and summarised in this paper.

Maybe the conclusions part could a bit more powerful with stronger messages on the next steps in this research area.

Reviewer #2: Overall this is a very comprehensive study with significant clinical and public health indications. The authors conducted a systematic review of the economic evaluation for STI treatment (gonorrhoea) by incorporating antimicrobial resistance (AMR), which is topical.

However, they have only identified a small number of studies. The majority of the studies are conducted either in the USA or UK. After 63 pages the only conclusion is “although ceftriaxone was the cheaper treatment, gentamicin became cost-neutral if the clinical efficacy of ceftriaxone reduced from 98% to 92%. I am unsure how these findings can be interpreted for low-middle-income countries particularly after accounting for the exchange rates.

Nevertheless given the limited information on this topic, I believe the current study will bring insight into the economic implications of AMR on treating STIs.

6. PLOS authors have the option to publish the peer review history of their article (what does this mean?). If published, this will include your full peer review and any attached files.

Reviewer #1: No

Reviewer #2: **Yes: **Handan Wand

---

## [Author Response · Author response to Decision Letter 0]

22 Jul 2023

We thank the reviewers for their comments and suggestions. We have revised the manuscript to reflect these and have provided specific responses below. 

Reviewer #1: Congratulations for selecting a research topic of great interest with results that may have potential implications for clinical practice and public health policies.

The systematic review methodology follows current evidence based practice, is well described and comprehensively documented in tables and supplementary files.

For PLOS ONE readers is highly relevant to be aware that there are no consistent methodological approaches in various economic analyses which could lead to variable results and thus recommendations. It is instructive to learn about the parameters influencing the results which are well described and summarised in this paper.

Maybe the conclusions part could a bit more powerful with stronger messages on the next steps in this research area.

Response: We thank the reviewer for their comment. We made conclusions based on the identified need for more developmental research to inform approaches to capture and incorporate AMR into economic evaluations. We apologise if this was not initially clear and have strengthened the conclusions as follows (lines 499-510) ‘There is dearth of data in relation to the economic analysis of AMR for curable STIs. However, by incorporating AMR costs (including in our exemplar study on alternative treatment for gonorrhoea) more robust interpretations in relation to the costs/cost effectiveness of new or alternative treatment and strategies can be made. The current evidence relating to the economic impact of AMR for curable STIs is generally limited to direct, aggregated costs over a short period, obtained from high income countries, is largely specific to gonorrhoea AMR, and often lacks a societal perspective. There is no standardised approach for the measurement of AMR in patients with curable STIs which results in uncertainty when reporting on the economic impact of AMR and makes comparisons between studies difficult. Further research is required to inform guidance on optimal approaches to capture AMR costs for curable STIs (e.g. how AMR is defined and how AMR associated costs are adequately captured) and methodically incorporate such costs into economic evaluations.’

Reviewer #2: Overall this is a very comprehensive study with significant clinical and public health indications. The authors conducted a systematic review of the economic evaluation for STI treatment (gonorrhoea) by incorporating antimicrobial resistance (AMR), which is topical.

However, they have only identified a small number of studies. The majority of the studies are conducted either in the USA or UK. After 63 pages the only conclusion is “although ceftriaxone was the cheaper treatment, gentamicin became cost-neutral if the clinical efficacy of ceftriaxone reduced from 98% to 92%. I am unsure how these findings can be interpreted for low-middle-income countries particularly after accounting for the exchange rates.

Nevertheless given the limited information on this topic, I believe the current study will bring insight into the economic implications of AMR on treating STIs.

Response: We thank the reviewer for their comment and agree that the current evidence base is limited as we have highlighted in lines 433-4434 (‘we found a small number of studies were eligible for inclusion, suggesting that the existing evidence base is limited’). This is why, in addition to the systematic review, we explored various approaches to incorporate AMR costs using prospectively collected data from a large RCT. Although most published evidence originated from high income countries as did our case-study, we discuss the likely implications for LMIC settings in relation to the cost of AMR (lines 437-440, ‘Cost estimates from the United States indicate that the cost of gonorrhoea resistance to current first-line treatment (ceftriaxone) is potentially substantial, equating to $235 to $378.2 million in direct medical costs over 10 years [36, 38]. These estimates are likely to be significantly higher in low and middle income settings [18]’). We also highlight the limitations of the review (lines 502-505, ‘The current evidence relating to the economic impact of AMR for curable STIs is generally limited to direct, aggregated costs over a short period, obtained from high income countries, is largely specific to gonorrhoea AMR, and often lacks a societal perspective’). We have added further comment in respect to our case study economic evaluation in lines 495-498, ‘Also, as our case study was conducted in a high income setting, the findings may not be generalizable to low income settings but contributes to the debate on how future economic evaluations could more fully incorporate the economic impact of AMR for curable STIs.

---

## [Decision Letter · Decision Letter 1]

18 Sep 2023

Economic evaluation of antimicrobial resistance in curable sexually transmitted infections; a systematic review and a case study

PONE-D-23-04588R1

Dear Dr. Ayinde,

We’re pleased to inform you that your manuscript has been judged scientifically suitable for publication and will be formally accepted for publication once it meets all outstanding technical requirements.

Kind regards,

Kwame Kumi Asare, Ph.D

Academic Editor

PLOS ONE

Additional Editor Comments (optional):

Reviewers' comments:

Reviewer's Responses to Questions

**Comments to the Author**

1. If the authors have adequately addressed your comments raised in a previous round of review and you feel that this manuscript is now acceptable for publication, you may indicate that here to bypass the “Comments to the Author” section, enter your conflict of interest statement in the “Confidential to Editor” section, and submit your "Accept" recommendation.

Reviewer #2: All comments have been addressed

2. Is the manuscript technically sound, and do the data support the conclusions?

Reviewer #2: Yes

3. Has the statistical analysis been performed appropriately and rigorously? 

Reviewer #2: Yes

4. Have the authors made all data underlying the findings in their manuscript fully available?

Reviewer #2: Yes

5. Is the manuscript presented in an intelligible fashion and written in standard English?

Reviewer #2: Yes

6. Review Comments to the Author

Reviewer #2: The manuscript technically sound, and do the data support the conclusions.

The authors made all data underlying the findings in their manuscript fully available

The authors adequately responded all my questions comments. No further comments.

7. PLOS authors have the option to publish the peer review history of their article (what does this mean?). If published, this will include your full peer review and any attached files.

Reviewer #2: **Yes: **Handan Wand

---

## [Editor Report · Acceptance letter]

11 Oct 2023

PONE-D-23-04588R1 

Economic evaluation of antimicrobial resistance in curable sexually transmitted infections; a systematic review and a case study 

Dear Dr. Ayinde:

I'm pleased to inform you that your manuscript has been deemed suitable for publication in PLOS ONE. Congratulations! Your manuscript is now with our production department. 

Kind regards, 

on behalf of

Dr. Kwame Kumi Asare 

Academic Editor

PLOS ONE